# OpenReview forum: "Benchmarking Anomaly Detection for Large Language Model Alignment"
_ICLR.cc/2026/Conference — Submitted to ICLR 2026_

### Official Review · Reviewer_NMsN · 2025-10-27

**Soundness:** 3
**Presentation:** 2
**Contribution:** 2
**Rating:** 2
**Confidence:** 3

**Summary:**

The paper propose to benchmarking the anomaly detection for LLM misalignment, specifically the  Benchmarking detection of unforeseen alignment failures. Thet  find that perplexity and Mahalanobis distance based detectors perform the best among these baselines, but no method performs at a high level across all failure modes.

**Strengths:**

1. Safety alignment is a very important question

2. Exploring which safety mechanisms are most effective among existing LLM alignment methods is a  timely research question.

3. The idea of adapting anomaly detection techniques from other domains to the context of LLM safety is interesting.

**Weaknesses:**

1.  Limited scope of anomaly detection in alignment.
The idea of using anomaly detection to safeguard LLMs is not yet a mainstream or widely adopted approach. Most prior works in this area use guard or moderation models directly for safety monitoring. The baselines used in this paper are primarily from non-LLM domains, which limits the significance of the benchmarking results. Since the proposed detector resembles guard models in nature, the research scope could be broadened to include a more direct comparison with established LLM alignment frameworks.

2. Unclear motivation compared to guard or moderation models.

Training an anomaly detector on preference datasets to assess input and output safety is conceptually similar to guard or moderation models (Llama Guard model), which are also trained to classify whether content is safe. The overall intuition and high-level purpose are essentially the same. Also, as the authors note, anomaly detectors tend to yield more false positives, which makes them appear less practical than guard models. Thus, it remains unclear why benchmarking anomaly detectors is necessary

3. Unfair model comparison.

In Table 2, different baselines are built upon different backbone models with varying sizes. Comparing them directly in one table may be misleading, as it is unclear whether the observed performance differences are due to the methodology or simply model capacity.

4. Limited model coverage.

As a benchmarking paper, the experiments would be more convincing if they covered a broader range of models and model sizes. Currently, the evaluation is restricted to Gamma, Llama-3.2, and a few smaller models (1B–8B), which limits the generality of the conclusions.

5. Limited insight or discovery.

A strong benchmark paper typically aims not only to compare methods but also to uncover meaningful patterns or insights that can guide future research. The current version focuses mainly on performance comparisons without offering deeper analysis, interpretations, or directions for follow-up work.

**Questions:**

Please see above

---

> ### Author Response · Authors · 2025-11-21
> **Responses to Reviewer NMsN**
>
> We thank the reviewer for their thorough review and helpful comments. Below, we have responded to weaknesses listed in the review.
>
> **"Limited scope of anomaly detection in alignment" and "unclear motivation compared to guard or moderation models"**
>
> We agree that guard/moderation models are the closest existing tool to the methods we study in this paper. However, guard models are trained to recognize known alignment failures such as jailbreaks, prompt injections, etc. This means that they often fail to recognize unforeseen alignment failures; we give a few examples of such unforeseen failures in the first paragraph of the introduction. Since unforeseen alignment failures are, by definition, outside a guard model’s training data, anomaly detection is the natural tool for detecting these unseen failures. Our benchmark, MAAD, aims to simulate real-world unforeseen failures by explicitly restricting the data that guard models can be trained on, and then testing them on examples of failures outside that training data. The reward model method that we test is quite similar to existing guard models; our reward models are trained on a dataset like HH-RLHF-Harmless to separate unsafe conversations (like those where the model gives unsafe information) from safe conversations (like those where the model refuses unsafe prompts). However, we find that this training does not generalize to unforeseen failure modes. This motivates the need for proper anomaly detection methods that can recognize anomalous alignment failures that are not in existing safety training data.
>
> **"Unfair model comparison" and "limited model coverage"**
>
> We are currently working to complete our evaluation across more model sizes for all methods and plan to have these full results ready if the paper is accepted.
>
> **"Limited insight or discovery."**
>
> Like many other benchmarks, we view our primary contribution as introducing a novel evaluation framework and dataset that enables progress on the important problem of misalignment anomaly detection. For example, benchmark papers like Humanity's Last Exam (Phan et al., 2025) focus on the process of constructing the benchmark and measuring baseline performance.

---

> > ### Comment · Reviewer_NMsN · 2025-11-25
> > **Reply to the rebuttal**
> >
> > Thanks to the authors for the response. However, I feel that my concerns have not been fully addressed, so I will keep my original score.

---

### Official Review · Reviewer_Twxe · 2025-10-29

**Soundness:** 3
**Presentation:** 3
**Contribution:** 3
**Rating:** 6
**Confidence:** 3

**Summary:**

This paper introduces MAAD (Mis-Alignment Anomaly Detection), a benchmark for anomaly detection targeted at alignment failures in large language models (LLMs). The core claim is that many real safety failures happen in unusual, previously unseen situations where models behave in unsafe, deceptive, or manipulative ways. The paper frames these situations as “unknown unknowns” in alignment: jailbreak prompts that bypass refusals, deceptive tool-use, extreme sycophancy, etc. MAAD is designed to test whether anomaly detectors can flag such situations without having ever seen that specific failure mode in their post-training data.

The authors evaluate several baseline anomaly detectors: (a) prompting an instruction-tuned model to self-rate how unusual a conversation is, (b) perplexity of the conversation, (c) reward models and ensembles of reward models, (d) Mahalanobis distance in hidden representations, and (e) “pessimistic” combinations of a reward model with the other signals. The main empirical result is that Mahalanobis distance and perplexity work best on average (mean AUROC 0.83 and 0.76), but no method is reliable across all seven failure cases, and all tested methods fail badly on at least one case (AUROC <= 0.7). The paper argues that this shows anomaly detection can act as a safety monitor for alignment failures, but current methods are still not reliable enough.

**Strengths:**

1. Clear problem formulation that focuses on alignment anomalies (normative failures like deception, sycophancy, unsafe compliance), not just factual uncertainty or hallucination.
2. MAAD enforces a “restricted post-training data” regime per failure case, which operationalizes the “unknown unknowns” story instead of assuming access to labeled examples of every safety failure.
3. Practical impact: provides an immediate evaluation target for scalable oversight systems and future safety monitors, and shows that current monitors are still far from reliable (no baseline is >0.7 AUROC on all cases).

**Weaknesses:**

1. The benchmark’s main claim depends on the “no leakage” guarantee that the allowed training data for a failure case does not already contain that failure mode (or close paraphrases), but this is not quantitatively audited in the main text. Without a leakage audit, it is hard to verify that detectors are truly detecting unseen failures rather than memorizing near-duplicates.
2. The paper merges two operationally different settings under one score: (a) anomalous prompts causing unsafe behavior (e.g., jailbreaks), where you would want a pre-generation gate, and (b) anomalous responses to normal prompts (e.g., sycophancy, controlling behavior), where you would want a post-generation filter. Averaging AUROC across both settings hides different practical requirements.

**Questions:**

For sycophancy and controlling behavior: are the malign conversations mined directly from real model outputs (e.g., GPT-4o sycophancy (Sharma et al., 2025a), Bing-like persuasive behavior (Roose, 2023)), or were they partially hand-edited / synthesized? If there was editing, what quality control was used to avoid stylistic artifacts that a detector could trivially exploit?

---

> ### Author Response · Authors · 2025-11-21
> **Responses to Reviewer Twxe**
>
> We thank the reviewer for their insightful comments; we are glad they found the paper had a "clear problem formulation" and "practical impact." Below, we have responded to specific comments from the review.
>
> **"For sycophancy and controlling behavior: are the malign conversations mined directly from real model outputs (e.g., GPT-4o sycophancy (Sharma et al., 2025a), Bing-like persuasive behavior (Roose, 2023)), or were they partially hand-edited / synthesized? If there was editing, what quality control was used to avoid stylistic artifacts that a detector could trivially exploit?**
>
> For the sycophancy dataset, we trained our own sycophantic model based on Llama-3.1-8B-Instruct by running iterative DPO using synthetically generated preference data. We constructed the preferences using a judge model that was instructed to choose prompts that agreed more with the user. For the controlling dataset, we system-prompted GPT-4.1-mini to respond in a controlling and misaligned manner. See Appendix B for all the details of how we constructed these datasets.
>
> Overall, we do not see major stylistic differences between the different datasets. Although we did not explicitly control for style, the fact that our baseline anomaly detection methods struggled on many of the datasets suggests that there are not trivially exploitable artifacts.
>
> **"The benchmark’s main claim depends on the “no leakage” guarantee that the allowed training data for a failure case does not already contain that failure mode (or close paraphrases), but this is not quantitatively audited in the main text. Without a leakage audit, it is hard to verify that detectors are truly detecting unseen failures rather than memorizing near-duplicates."**
>
> We agree that a quantitative audit is necessary to substantiate our claim of minimal leakage. The MAAD benchmark attempts to mitigate the leakage challenge by artificially constraining the post-training data, but a full quantitative audit of the degree of overlap remains challenging.  We are exploring a few strategies to audit the training data to ensure minimal leakage and will post the results here if we finish before the end of the discussion period.
>
> **"The paper merges two operationally different settings under one score: (a) anomalous prompts causing unsafe behavior (e.g., jailbreaks), where you would want a pre-generation gate, and (b) anomalous responses to normal prompts (e.g., sycophancy, controlling behavior), where you would want a post-generation filter. Averaging AUROC across both settings hides different practical requirements."**
>
> We thank the reviewer for their suggestion. We plan to separate out the AUROC scores for these two settings and more thoroughly discuss the distinction between them if the paper is accepted.

---

> > ### Comment · Reviewer_Twxe · 2025-11-26
> >
> > I thank the authors for their efforts to address the raised points. And I keep my original score.

---

### Official Review · Reviewer_NhuH · 2025-10-29

**Soundness:** 2
**Presentation:** 1
**Contribution:** 2
**Rating:** 2
**Confidence:** 5

**Summary:**

In this paper, the authors use anomaly detection as a safety net. In other words, they build methods that can monitor a deployed LLM (i.e., prompt‐response pairs) and detect when something looks anomalous (i.e., very unlike what was seen in training) and thus potentially misaligned, even if the exact failure mode was never seen in training. To this end, they introduced MAAD - a benchmark for evaluating anomaly detection methods for LLM alignment. After evaluation, they found that none of them uniformly succeed across all failure modes, but certain methods (like Mahalanobis distance and perplexity) do relatively well on average.

**Strengths:**

- The paper is complete

**Weaknesses:**

- The **challenge and motivation** for this project remain somewhat unclear to me. Specifically, I am not fully convinced by the claim that *“anomaly detection is the key tool for preventing safety failures of LLMs in unusual situations.”* The evaluation results show that existing anomaly detection methods perform poorly, so it is unclear how they can serve as the key tool for safety prevention. More insights and justification are needed to clarify this argument.

- As I understand it, the paper reframes existing *safe/unsafe* categorizations under the concept of *anomaly detection*. However, I find the necessity and novelty of this framing insufficiently explained.

- Moreover, the **writing quality** requires significant improvement. Many sentences are vague and difficult to interpret. For example:
  > “Our key insight is that we can force certain known alignment failure modes to remain unseen by explicitly restricting the post-training data that anomaly detection methods can use within MAAD.”
  This sentence is particularly unclear and would benefit from clearer wording and explanation.

- For **Sections 3.1.1 and 3.1.2**, it would be helpful to explain why these specific failure cases were chosen. Any underlying rationale or insights behind the selection would strengthen the paper.

- There exist **many more anomaly detection methods** than those used as baselines in the experiments—for instance, *Self-Inf* [1]. The choice of baselines should be better justified, especially in light of the diversity of existing approaches.


[1] Pruthi G, Liu F, Kale S, Sundararajan M. Estimating training data influence by tracing gradient descent. Advances in Neural Information Processing Systems. 2020;33:19920-30.

**Questions:**

See above

---

> ### Author Response · Authors · 2025-11-21
> **Responses to Reviewer NhuH**
>
> We thank the reviewer for providing helpful comments to improve the quality of the submission. Below, we have responded to individual points in the review.
>
> **"The evaluation results show that existing anomaly detection methods perform poorly, so it is unclear how they can serve as the key tool for safety prevention."**
>
> The reviewer states that our results showing that existing anomaly detection methods perform poorly invalidate our claim that anomaly detection is a key tool for preventing safety failures. However, this claim is meant to be motivation for working on the problem of anomaly detection, and is supported by the first two paragraphs of the introduction. In particular, we argue that other alignment techniques, like safety training or input/output filters, often fail to generalize to unforeseen alignment failures. This has been extensively demonstrated by some of the related work we cite in the introduction. Thus, anomaly detection may be the only way to reliably prevent these unpredictable failure modes. Although our later results suggest that the problem is hard, we believe that our contribution of a novel evaluation approach and benchmark for anomaly detectors will enable more progress on this important problem.
>
> **"The paper reframes existing safe/unsafe categorizations under the concept of anomaly detection."**
>
> Our work highlights a fundamental distinction between traditional safe/unsafe classification methods (e.g., moderation/guard models) and the problem of anomaly detection targeted by the MAAD benchmarkBench. Past tools for monitoring/filtering LLMs to prevent alignment failures (e.g., constitutional classifiers, Llama Prompt Guard, gpt-oss-safeguard) depend on explicitly labeled examples of unsafe inputs included in their training data. In contrast, MAAD specifically assesses a model’s capability to identify potentially unsafe inputs from unusual distributions that were absent or underrepresented during training. This is a significantly different and more challenging task, as traditional classification approaches to moderation, trained broadly across many domains, may struggle to recognize alignment failures from outside their training distribution. With MAAD, we present a novel evaluation framework in which we intentionally restrict the available training data for anomaly detection models, ensuring that certain failure modes are completely absent during training. As we show, many existing methods struggle in this setting, demonstrating the importance of further work on misalignment anomaly detection.
>
> **"There exist many more anomaly detection methods than those used as baselines in the experiments—for instance, Self-Inf [1]. The choice of baselines should be better justified, especially in light of the diversity of existing approaches."**
>
> We appreciate the reviewer's suggestion to include methods like Self-Inf (Pruthi et al., 2020). However, our current set of baselines focuses exclusively on methods that are viable for post-hoc monitoring in a deployed LLM setting. Self-Inf requires computing influence scores on several training checkpoints, making it infeasibly costly to use for monitoring. Overall, our chosen baselines are chosen to be representative of general categories of anomaly detection methods—ensemble-, perplexity, and latent space-based anomaly detection. However, we will work to cite more methods such as Self-Inf in our related work and explain why they may be inapplicable to our setting.

---

### Official Review · Reviewer_QquG · 2025-10-31

**Soundness:** 1
**Presentation:** 2
**Contribution:** 1
**Rating:** 2
**Confidence:** 5

**Summary:**

This paper introduces MAAD, a new benchmark for evaluating the ability of anomaly detection methods to identify unforeseen alignment failures in LLMs. The central premise is that as LLMs are deployed, they will encounter "unknown unknown" failure modes not covered by their safety training. Anomaly detection is proposed as a key tool to mitigate these risks. The core design of MAAD is to test whether detectors can identify known types of alignment failures (e.g., jailbreaks, sycophancy, tool-call deception) when they have been explicitly excluded from the detector's training data. This setup simulates the challenge of detecting novel failure modes. The benchmark comprises seven distinct failure cases, each with curated training and test sets. The authors evaluate several baseline anomaly detection methods on MAAD, including prompting-based approaches, perplexity, reward models, and Mahalanobis distance on internal representations. The results show that while perplexity and Mahalanobis distance-based methods perform best, no single method achieves high performance across all failure modes, motivating the need for further research in this area.

**Strengths:**

The paper addresses a critical and timely problem in LLM safety that how to guard against unforeseen alignment failures. The argument that anomaly detection can serve as a crucial safety net for "unknown unknowns" is compelling and relevant to the broader goal of deploying trustworthy AI systems.

**Weaknesses:**

1.The paper successfully demonstrates that existing anomaly detection methods perform poorly on the MAAD benchmark, but it stops short of providing a deep analysis of why they fail. For instance, why does Mahalanobis distance excel at detecting sycophancy but struggle with function-calling deception? A more detailed investigation into the relationship between failure modes and the shortcomings of specific detection methods is needed. Furthermore, the paper does not propose any new methods or concrete paths forward to address the identified gaps, limiting its impact.

2.The main body of the paper is severely lacking in experimental detail and results. It contains only one primary results table (Table 2), which presents a high-level summary of AUROC scores. Much of the crucial experimental detail, such as the analysis of scaling trends and more granular results, is relegated to the appendix. This makes it difficult for the reader to fully assess the experimental findings without constantly referencing supplementary materials.

3.The primary contribution of the paper is the collection and reorganization of existing, known LLM failure datasets into a new benchmark structure. While this is a valuable engineering effort, the paper does not introduce any new types of alignment failures or provide fundamentally new insights into the nature of LLM misalignment itself. The work is more of a problem statement and dataset curation than a novel research discovery.

**Questions:**

1.The results show that no single baseline performs well across all failure modes. Does this suggest that a universal anomaly detector for LLM alignment is infeasible, and that future work should instead focus on ensembles of specialized detectors? What are the authors' thoughts on the path forward, beyond simply stating that "further work is needed"?

2.The Mahalanobis distance method shows surprisingly poor scaling, with performance degrading as model size increases. The paper hypothesizes that this is due to noise from higher-dimensional representations. Have the authors considered or tested dimensionality reduction techniques (e.g., PCA) on the hidden states before calculating the distance to see if this mitigates the issue?

3.The paper's premise is that anomaly detectors can identify "unforeseen" failures. However, the benchmark is constructed from known failure types. How confident are the authors that strong performance on MAAD would translate to effective detection of truly novel, undiscovered failure modes in the wild? Is there a risk that methods developed to succeed on MAAD might simply be better at generalizing across known categories of failure, rather than detecting genuinely new patterns?

---

> ### Author Response · Authors · 2025-11-21
> **Responses to Reviewer QquG**
>
> We thank the reviewer for their thorough review and helpful comments. Below, we have responded to individual weaknesses and questions in the review.
>
> **"The primary contribution of the paper is the collection and reorganization of existing, known LLM failure datasets into a new benchmark structure."**
>
> While our benchmark does incorporate some existing datasets, our key contribution lies in the novel evaluation framework we introduce for assessing misalignment anomaly detection in LLMs—a crucial but understudied aspect of scalable oversight. In our benchmark, MAAD, we propose the methodology of explicitly limiting the post-training data anomaly detectors can use. This ensures that our test-set anomalies are truly out-of-distribution, and to our knowledge, this is a novel setting for evaluating LLM-based anomaly detection. Our approach enables empirical measurement of anomaly detection capability, which has previously been challenging because modern models post-train on extensive corpora, complicating the task of defining true anomalies.
>
> **"Does this suggest that a universal anomaly detector for LLM alignment is infeasible, and that future work should instead focus on ensembles of specialized detectors?"**
>
> The methods we test are common baselines taken from the literature, such as uncertainty calibration and outlier detection. Including these baselines was not motivated by demonstrating that they would perform well at anomaly detection, but rather the opposite: –that existing techniques may not be sufficiently strong at anomaly detection. Our aim is to foster further work in this area by providing a carefully constructed evaluation for anomaly detectors, and evaluating a few existing baselines—similarly to many other benchmarks. For example, benchmarks such as HLE, ARC-AGI, and StrongREJECT were created to drive progress on important problems by demonstrating the shortcomings of existing methods. We see MAAD as fulfilling a similar purpose for misalignment anomaly detection. This does not imply that a universal anomaly detector is infeasible—just that more work on this important problem is necessary.
>
> **"Have the authors considered or tested dimensionality reduction techniques (e.g., PCA) on the hidden states before calculating the distance to see if this mitigates the issue?"**
>
> We thank the reviewer for their insightful suggestion. We are working on this and will post results if we have them before the end of the discussion period.
>
> **"How confident are the authors that strong performance on MAAD would translate to effective detection of truly novel, undiscovered failure modes in the wild?"**
>
> The MAAD benchmark intentionally provides models with a limited set of post-training data, substantially less comprehensive than the extensive post-training modern LLMs typically undergo. Consequently, although our evaluation focuses on known failure modes within a constrained set of scenarios, these inputs effectively represent "novel, undiscovered failure modes" from the viewpoint of a model with such restricted post-training. Therefore, we believe that methods demonstrating strong generalization across the diverse and numerous scenarios in MAAD are likely to be effective in detecting genuinely novel failures in real-world applications. Furthermore, we intend to continually expand the MAAD benchmark by incorporating additional settings to further improve its comprehensiveness.

---

### Meta-Review · Area_Chair_7Qaw · 2026-01-03

**Summary:**

This paper designed a new benchmark to evaluate the ability of anomaly detection methods in finding alignment failures of LLMs. The motivation is that anomaly detection is a promising tool to mitigate the failure modes caused by unknown unknowns. The goal of this benchmark is to motivate anomaly detection as an approach to LLM safety.

**Reviewer Concerns:**

The problem studied in this paper is clearly formulated, interesting, and timely. The paper demonstrates that current monitors are still not very effective. The concerns about the submission are that (1) insight or findings made in this benchmark paper are limited, i.e., the paper finds an important problem but lacks interesting research contributions, (2) the experiments can be further enhanced, such as experimental details and more evaluation settings.

**Reviewer Scores:**

It is very likely that the reviewers would not significantly change their scores.

---

### Decision · Program_Chairs · 2026-01-26

Reject